# Does It Run in the Family? How Family Background Affects Attachment Styles for Students in Higher Education

**DOI:** 10.3390/ijerph18105135

**Published:** 2021-05-12

**Authors:** Bent E. Mikkelsen, Anette Q. Romani, Inger G. Bo, Frantisek Sudzina, Maria P. Brandão

**Affiliations:** 1Department of Geosciences and Natural Resource Management, University of Copenhagen, Rolighedsvej 23, DK-1958 Frederiksberg, Denmark; bemi@ign.ku.dk; 2Department of Sociology and Social Work, Aalborg University, Fibigerstræde 13, 122, DK-9220 Aalborg, Denmark; aqr@socsci.aau.dk (A.Q.R.); inger@socsci.aau.dk (I.G.B.); 3Department of Materials and Production, Faculty of Engineering and Science, Aalborg University, A. C. Meyers Vænge 15, 2450 Copenhagen, Denmark; sudzina@business.aau.dk or; 4Department of Systems Analysis, Faculty of Informatics and Statistics, University of Economics, nám. W. Churchilla 1938/4, 130 67 Prague, Czech Republic; 5School of Health, University of Aveiro, Edifício 30, Agras do Crasto-Campus Universitário de Santiago, 3810-193 Aveiro, Portugal; 6Centre for Health Technology and Services Research, University of Porto, 4200-450 Porto, Portugal

**Keywords:** social attachment, universities, students, family

## Abstract

Background: Socioeconomic background has traditionally been the most important determinant of an individual’s social advantage. Studies have used social class and opportunities based on parental income and education to predict such advantage. There is limited evidence that stratification mechanisms other than socioeconomic background can play an important role. The purpose of the study is to examine the influence of the traditional factors (income and education) of family background on students’ social attachment styles compared to other background variables (civil status and number of children). Methods: We used the Vulnerable Attachment Style Questionnaire as an outcome measure to assess students’ social attachment advantage. As a point of departure, we use theories of social psychology to categorize social relations in terms of secure or insecure bonding, respectively. Results: A cross-sectional data set of 912 university students from five European countries was used. With respect to social attachment, the likelihood of being a student with robust relations increases by 23% if the students have high-income parents. Students with robust relations also have a decreased likelihood of poor body self-esteem by 19% when compared with other students. Conclusions: Stratification mechanisms other than social class, such as parental characteristics, civil status, and number of siblings, all affect the privileged students’ social relations.

## 1. Introduction

Young people today live in an era of late modernity that is characterized by a high degree of individualism and reflexivity [1,2,3]. Thus, this notion of reflexivity helps to explain how young people change their position in the social system of which they are a part and shape their own identity through the definition, for example, of norms, preferences, likes, dislikes and attitudes. At the same time, social interaction patterns of young people have undergone a substantial transformation over recent decades, fueled by the emergence of social media.

Communicating through social media is radically different from what has been experienced earlier, as face-to-face interaction has sometimes been replaced by faceless interactions on platforms such as Facebook, Twitter, Instagram and Snapchat. In social interactions, students can establish secure bonds if they have balanced relations that are neither too dependent nor too independent of other people. On the other hand, students can have insecure bonds if they have unbalanced relations where they are either too dependent and engulfed or too independent and isolated [4] or they may even have violent tendencies [5]. It can be speculated that, with exposure to social media, the importance of bonds via face-to-face interactions among young people can be reinforced. Thus, it is important for young people’s well-being to have secure bonds in order to avoid being under-socialized or over-socialized. But what is shaping the social interaction patterns of young people? According to Bourdieu [6,7] and recent research [1,8], high parental education and income have a beneficial impact on social relations, and, as a result, considering socioeconomic factors in family background has traditionally been the main focus of studies on patterns of social attachment.

The purpose of this study is to examine the influence of family background factors on students’ social attachment styles compared to other background variables. Thus, we go beyond considering the traditional factors related to social class, such as parental income and education, by considering variables that can be related to late modernity, such as civil status and number of children. 

### 1.1. State of the Art

The study of social attachment is concerned with how individuals interact socially [9,10]. Impaired social relations can manifest themselves as social anxiety and social avoidance [11,12], where social anxiety is associated with fear of embarrassment and humiliation in social interactions. Individuals with high attachment anxiety can show proximity-seeking behavior, which can be overinvolved and controlling. In contrast, social avoidance implies that individuals tend to avoid closeness and reliance on other individuals and emphasize self-reliance and independence, as they either deny or are overoptimistic about their own ability to handle distress [9]. 

Impaired social relations have been reported to manifest as either secure or insecure social attachment [13]. Secure individuals, according to this approach, are believed to have optimistic beliefs about their ability to handle distress and to accept negative aspects of themselves [14]. Secure individuals believe that others are reliable and can be trusted, and they do not hesitate in asking for support if needed [15]. On the contrary, insecurely attached individuals—according to this way of thinking—have low self-esteem and poor social skills, and suffer frequently from loneliness and interrelated problems [16,17]. Compared to the literature mentioned above, our theoretical perspective using Scheff [18] is interactional, implying that our focus is on social interaction rather than on the individual. We expand this approach by claiming that variations in attachment style can be considered as habitually determined [19] and as internalized experiences with significant others [10]. 

### 1.2. Theoretical and Conceptual Foundation 

According to Bourdieu [1,6,7], social heritage can be either direct, in the form of a social network, or understood indirectly as the social competencies which the student has been socialized into throughout their childhood. Nevertheless, in late modernity, socioeconomic background, as a predictor of outcomes, seems to be challenged by other factors. According to Giddens, individual actions need not be extensively thought about in pre-modern societies, because available choices are already determined by social class and traditions. In contrast, in late modernity, individuals are much less concerned with inherited traditions. Thus, individuals have more choices [2], implying that individual actions and decisions now require more analysis and reflection [20,21]. We claim that nowadays both mechanisms might be at stake. 

For Giddens [20,21], late modernity has created new patterns of self-identity. Individuals are increasingly concerned about individualization and oriented towards self-realization. This disappearance of traditional values and norms is one important feature of late modernity. People marry and remarry, which can be seen in high divorce rates. As a result, family ties are ever-changing [20,21]. This creates highly fluid family environments where young people have to navigate a patchwork of numerous different kinds of social relations. Beck and Beck-Gernsheim [22] suggest that as a consequence of secularization children increasingly are considered as the stable foundation in the lives of their parents. It can be speculated that the lack of stability in parental relationships causes a growing desire for perfection in the life of their child. 

In late modernity, religion is no longer the foundation of society. If people live religious lives, it is a reflective choice rather than the common tradition in society. This secularizing implies that the love for the child gives meaning to parents’ lives and thus replaces religion [22]. This transition puts big pressure on a child and might increase the risk of more egocentric children who lack the full ability to obtain and maintain robust relations.

Social attachment captures how people interact socially and whether they are uncomfortable having people close or are anxious about being left alone [10]. Social attachment can be understood as a social bond, and this bond can be either secure or insecure [4,18,23]. A secure bond can be described as a balanced interaction in which the interactors are not too distant or too close. As described by Scheff [18,23,24], the bond can also be insecure. The secure bond is shared by individuals who are not emotionally too distant or too engulfed, for example, students who are able to stand on their own feet and at the same time are able to engage with other people. The person manifesting an insecure bond can be either engulfed, implying that the individual is over-engaged in social relations, or isolated, implying that the individual is under-engaged in social relations [4,18,23]. For example, engulfed students are very dependent on group attitude and behavior, whereas isolated students are unable to rely on others and therefore are unable to fit into the group. 

According to Scheff, individuals’ relations can be seen on a continuum ranking from engulfed towards isolated. He claims that a balanced bond is a psychosocially healthy relationship; meanwhile, he considers over- and under-engaged relations as psychosocially unhealthy [4,18,23]. How can such a continuum be illustrated? We define it by labeling students at three different positions: engulfed, robust and isolated. In our data set, we have 454 students with robust relations, 155 students with engulfed relations and 178 students with isolated relations. 

## 2. Materials and Methods

### 2.1. Participants

The authors have been part of one of the working groups of the European Cooperation in Science and Technology (COST) Action: IS1210-Appearance Matters: Tackling the Physical and Psychosocial Consequences of Dissatisfaction with Appearance. This study is part of a larger study that was designed in that working group. The total population included 980 students from five European countries: Denmark, Germany, Portugal, Croatia, and the Czech Republic. In each country, the students were selected by convenience method [25]. The details of the criteria for including students and how they applied to each country, as well as other information about methodological steps taken, are closely detailed elsewhere [26]. This study was performed in accordance with the Helsinki Declaration and with local ethical guidelines. In Portugal, the study was endorsed by the Ethics and Deontology Committee of the University of Aveiro (Process 5 to 9/2016). Before data collection, all participants were informed about the study and signed an informed consent form. However, in this article considering social attachment in VASQ, we have information on 912 students. Thus, we are missing information for 6.08% on social attachments. The missing information is mainly from students from Germany. In contrast, for the Czech Republic, we have no missing information on any of the students. As our missing information is mainly driven by Germany, where the sample is very large, it will most likely not bias the results substantially.

### 2.2. Instruments

The data set, used to investigate the association between more highly educated students, poor body self-esteem and poor social attachment, is part of the COST action 1210 Appearance Matters. The survey data were collected in higher education facilities during 2016 by members of a working group. The data collection was done via the educational programs with which the COST action members were affiliated. 

### 2.3. Procedure

The dependent variable to capture social relations was constructed based on a validated Vulnerable Attachment Style Questionnaire (VASQ). The VASQ includes 22 items and is a 5-point Likert scale on which respondents rate their degree of agreement with each statement. Response options range from ‘strongly agree’ to ‘strongly disagree’. Using dummy variables, we divided the attachment style into robust, engulfed and isolated social relations with a Cronbach’s alpha of 0.8336, 0.685 and 0.8379, respectively. 

The independent variable was parental background characteristics. Thus, we had information on students, including gender, firstborn, number of siblings, parental employment, parental income, maternal education and civil status of parents. According to the analysis, the only variables that matter are siblings, parental civil status and income, where number of siblings are coded into no siblings or some siblings. Parental income captures yearly net income above the country-specific median income, which we coded into high and low parental income. Civil status of parental was coded to capture living with one or two biological parents. All variables were coded into dichotomy variables to avoid having very few students in each category. 

Setting (Country). 

The data were collected in five countries, namely Denmark, Croatia, Portugal, the Czech Republic, and Germany. In this study, we had 163 students from Denmark, 105 students from Croatia, 157 students from Portugal, 299 students from the Czech Republic and 256 students from Germany. However, considering those students who did not answer the questions on VASQ, we had 912 students.

Field of study. 

The sample was divided according to field of study into Humanities and Social Sciences (arts and humanities, social sciences, and management) and Nature and Technical Sciences (engineering and technology, life sciences and medicine, natural sciences); 571 students were studying natural and technical sciences and 341 students were studying humanities and social sciences. Among those studying humanities and social sciences, 232 students had robust relations, 47 students had engulfed*,* and 52 students had isolated relations. Among those studying natural and technical sciences 261 students had robust relations, 125 students had engulfed, and 145 students had isolated relations. We underline that 50 students could not be categorized, so we placed them simultaneously place them in both fields of study. 

According to relationship, we divided the students into secure—robust—and insecure further divided into engulfed or isolated.

### 2.4. Data Analysis

When analyzing the relationship between different students’ relations (robust, engulfed and isolated), we applied a simple least square regression (OLS). Both our dependent and independent variable were dummy variables. We used a linear probability model when making the regressions. For ease of interpretation, we used OLS to estimate the binary model rather than a binary logistical regression. The OLS in the Linear Probability Model (LPM) are consistent estimates of the average probability derivatives, but the standard errors are biased because of heteroskedasticity [27,28,29]. When including robust errors to control for heteroskedasticity, standard errors barely change. We have estimated probit models for all our outcomes, and the results are substantially similar. Overall, our results from the linear probability model are a conservative estimate (results can be sent upon request). STATA 16 was used to perform the analyses. 

We use the following equation: *R*_i_ = 𝛽_0_ + 𝛽_1_ *Z*_i_ + *ε*_i_(1)
where *R* captures the relations of students *i*. *Z* is the student and parental control variable. *ε*_i_ is the error term. 

We argue that the students’ relations may result in internal or external behavior on the part of the students.
*B*_i_ = 𝛽_0_ + 𝛽*R*_1_ + 𝛽_2_ *Z*_2_ + *ε*_i_(2)
where *B* captures the likelihood of internal or external behavior of student *i*. *R* captures the relations of student *i*. *Z* is the student and parental control variable. *ε*_i_ is the error term. 

All tables present the beta coefficient from the regressions and the standard error. The statistical power of the result has been reported using * *p* < 0.05, ** *p* < 0.01, *** *p* < 0.001 level. We only report results that were statistically significant.

## 3. Results

We assume that social attachment is affected by trends in society which manifest themselves according to the different family characteristics presented in Table 1. To explore this issue, we consider whether parental characteristics vary according to students’ robust, engulfed and isolated relations. 

With regard to social attachment, our results presented in Table 2 indicate that students with engulfed relations vary significantly from students with robust relations. Thus, students with engulfed relations to a larger extent live with two parents and no siblings, and to a lesser extent have high-income parents. More specifically, 86.45 percent of the students with engulfed relations live with two parents compared to 66.74 percent of the students with robust relations. Furthermore, 16.77 percent of the students with engulfed relations have no siblings compared to 9.03 percent of the students with robust relations. Finally, 41.83 percent of the students with engulfed relations have high-income parents compared to 77.33 percent of the students with robust social relations.

In comparing students with isolated relations to those with robust relations, we find that students with isolated relations to a larger extent have no sibling and to a lesser extent have parents with a high income. In fact, 14.60 percent of the students with isolated relations have no siblings compared to 9.03 percent of the students with robust relations. Further, 52.89 percent of the students with isolated relations have parents with a high income compared to 77.33 percent of the students with robust relations. 

As shown in Table 3 with respect to social attachment, in column (1), we find that the likelihood of being a student with robust relations decreases by 13.33 percentage points if the student lives with both parents and by 12.06 percentage points if the student has no siblings. Likewise, we find that the likelihood of being a student with robust relations increases by 23.41 percentage points if the student has high-income parents. In column (2), the results indicate that the likelihood of being a student with engulfed relations increases by 13.32 percentage points if the student lives with both parents and by 6.96 percentage points if the student has no siblings. Likewise, we find that the likelihood of being a student with engulfed relations decreases by 16.62 percentage points if the student has high-income parents. In column (3), the results indicate that the likelihood of being a student with isolated relations increases by 6.45 percentage points. Likewise, we find that the likelihood of being a student with isolated relations decreases by 11.93 percentage points if the student has high-income parents.

Table 4 captures the relationship between attachment style (robust, engulfed and isolated relations) and two different reactions to attachment style—internally and externally oriented, respectively. While internal orientation is captured by body self-esteem, external orientation is captured by binge drinking. Column (1) shows that students with robust relations have a decreased likelihood of poor body self-esteem by 19.02 percentage points, but there is no significant impact on binge drinking. Column (2) indicates that students with engulfed relations have an increased likelihood of binge drinking by 9.94 percentage points, but with no significant impact on poor body self-esteem. Column (3) shows that students with isolated relations have an increased likelihood of poor body self-esteem by 9.93 percentage points, but there is no significant impact on binge drinking.

## 4. Discussion

Our results indicate that family characteristics other than traditional social class affect students’ social relations. Thus, the likelihood of students obtaining robust social relations seems to decrease if the students live with both parents and have no siblings but increases if the students have high-income parents. In contrast, the likelihood of students having either engulfed or isolated relations increases if the students live with both parents and have no siblings but decreases if the students have high-income parents. Perhaps we can think that the robustness of social relations can be influenced by the happiness that each person feels. The happier we are, the more able we are to create bonds. A very recent study shows that an increase in income leads to an increase in happiness, and individualism is associated with a lower power distance [2].

In line with Giddens [21], it can be speculated that the increased prevalence of poor social attachment is a result of trends in late modernity. We claim that people living in late modernity are individualized and oriented towards self-realization, and therefore family ties are very changeable. This creates an unstable climate where the young must be capable of navigating many kinds of social relations. Such a trend affects the likelihood of having more fragile and unstable family ties. This might generate children who in general are more robust or more insecure. Scientific evidence has shown that those who live in a family environment with better economic well-being are happier and more satisfied [30].

Parental self-realization and complex families can explain why living with both parents decreases the likelihood of students obtaining robust social relations and increases the likelihood of having engulfed or isolated relations. The increased individualization in late modernity creates more fragile relations because every social relation, even those among parents, can be broken. There is only one bond left which will never be torn apart and that is the bonding to the child. Crucial in this respect is the tendency to put too much attention on the child—the so-called project child. Beck and Beck-Gernsheim [22] argue that more emphasis is put on the child and much energy is invested in creating “the perfect child” as a kind of secularized religion. This means that bonding to the child is the meaning of life and replaces religious worship for the parents. That creates a lot of pressure on the child and disrupts the child’s self-understanding; they become egocentric and see themselves as the center of the world. Thus, being the only child decreases the likelihood of students obtaining robust social relations and increases the likelihood of having engulfed or isolated relations. 

Bourdieu’s concept, corroborated by Farrugia et al., is that economic capital still has a say with regard to young students’ social relations [1,6,7,8]. Economic capital can be transformed into social capital which can explain why high parental income increases the likelihood of students obtaining robust social relations and decreases the likelihood of having engulfed or isolated relations. We likewise find that students with robust and isolated social relations tend to be internally oriented or, more specifically, have poor body self-esteem. In contrast, students with engulfed relations tend to be externally oriented or, more specifically, have a higher risk of being binge drinkers. 

Taking a social psychological theoretical perspective on more highly educated students’ social relations as a point of departure, this paper has investigated the relationships between more highly educated students’ family background and their social relations, using cross-country data from 912 university students in Denmark, Germany, Portugal, Croatia and the Czech Republic. In line, with Scheff [4,24], we have categorized students’ relations into secure and insecure relations, where insecure relations can further be divided into engulfed and isolated relations.

We find that other stratification mechanisms than those traditionally claimed by Bourdieu [1,6,7] have been put into play. According to Giddens, people are much more individualized and occupied with self-realization, and therefore traditional family values, such as solidarity and family bonds, are no longer at stake [21]. Therefore, the family is much more fragile, and the family can easily fall apart. This creates an unstable climate where the children must cope with changeable family relations. Parental self-realization and changeable family relations can help us grasp why living with both parents decreases the likelihood of students obtaining robust social relations and increases the likelihood of having engulfed or isolated relations. Beck and Beck-Gernsheim [22] argue that parental love for the child can be understood as a secularized religion. This tendency puts a lot of pressure on the child, implying that being the only child decreases the likelihood of students obtaining robust social relations and increases the likelihood of having engulfed or isolated relations.

Not surprisingly, Bourdieu still has a say in explaining why high parental income increases the likelihood of students obtaining robust social relations and decreases the likelihood of having engulfed or isolated relations. Finally, our results reveal that students with robust and isolated social relations tend to be internally oriented, whereas students with engulfed relations tend to be externally oriented.

### Limitations

A study such as this obviously has some limitations. The results from the current paper are subject to several interpretations, which highlight the importance of future work in this era. The major critical point is the endogeneity problem. Another is the omitted variable bias, as we only consider observable characteristics of student and parent. For example, children with no siblings may differ from children with many siblings in ways that we do not observe, suggesting that parental ambitions among others may be driving our results. Further research in this field should continue to address secure and insecure relations attempting to disentangle the true underlying mechanisms. Another critical point is the causality direction, e.g., is it its insecure relations that affect internal and external orientation or is it internal and external orientation that affect insecure relations? Further research should look for exogenous variation in students’ social relations in order to determine the causality direction. Regardless of these limitations, our findings add new evidence to the debate on young people’s social bonds as we find that not only social class but also family ties, which can promote social skills, are important.

## 5. Conclusions

According to various researchers [6,7,8], economic, cultural and social capital can be captured by the concept of symbolic capital, thus implying that currently individuals experience more life changes and can occupy a privileged position in society. Our findings point towards other stratification mechanisms that are important in late modernity. More specifically, social skills are extremely important, and these are dependent on family background, whether a person lives in a one-child family and whether one grows up in a family with two parents. The specific topics of our conclusions are:Increases in family income contribute to students having robust relations.The existence of family ties contributes to increasing student’s social skills.Students with engulfed relations have an increased likelihood of drinking and are therefore potentially less healthy.

## Figures and Tables

**Table 1 ijerph-18-05135-t001:** Description of sociodemographic characteristics of sample ^(i)^.

Outcome	Question	Outcome Range	Mean (Std)
High income	What was your parent’s yearly net income in the last calendar year including wages, salaries, self-employment and any other sources of income including transfer payments such as unemployment benefit or pension)? More than (median income) equal 1, 0 otherwise	0–1	0.6624 (0.4832)
High education	What is the highest level of education your mother has completed? University or college or equivalent equal 1, 0 otherwise	0–1	0.2945 (0.4560)
Fulltime working	Which of the following statements about occupation status apply to your parents? Full-time work equal 1, 0 otherwise	0–1	0.8166 (0.3871)
Two parents	Select the current marital status of your biological parents? Married/unmarried and both parents living together equal 1, 0 otherwise	0–1	0.7270 (0.4456)
Female	What is our gender? Dummy variable =1 if the students are reply female, 0 otherwise	0–1	0.6220 (0.4851)
No siblings	Do you have any sibling, if yes how many? Dummy variable =1 if the students reply none, 0 otherwise	0–1	0.1215 (0.3269)
One sibling	Do you have any sibling, if yes how many? Dummy variable =1 if the students reply one, 0 otherwise	0–1	0.4757 (0.4996)
Firstborn	Dummy variable =1 if the students have younger sisters or brothers, 0 otherwise	0–1	0.5832 (0.4994)

(i): *n* = 912 for all variables. *n* = 809 for high income as there is not income information from Croatia. Present means for the sample. Standard deviations in parentheses.

**Table 2 ijerph-18-05135-t002:** Description of types of student and family background.

	Engulfed	Robust	Isolated
Two parents	0.8645 * (0.3433)	0.6674 (0.4716)	0.8146 (0.3897)
No siblings	0.1677 *** (0.3748)	0.0903 (0.2869)	0.1460 * (0.3541)
High income	0.4183 *** (0.4958)	0.7733 (0.4191)	0.5289 (0.5012)
***n***	155	454	178

Note: *n* = 155 for student with engulfed relation, *n*= 454 for students with robust relations and *n* = 178 for students with isolated relation. For high income, *n*= 98 for student with engulfed relation, *n*= 450 for students with robust relations and *n* = 176 for students with isolated relations, as there is not income information from Croatia; * *p* < 0.05, *** *p* < 0.001. Present means for the different types of students. Standard deviations in parentheses.

**Table 3 ijerph-18-05135-t003:** Relation between family background and students’ relations.

	Secure	Insecure	
	Robust	Engulfed	Isolated
Two parents	−0.1333 *** (0.0388)	0.1332 *** (0.0268)	0.0645 ** (0.0301)
No siblings	−0.1260 ** (0.0545)	0.0696 * (0.0376)	0.0365 (0.0423)
High income	0.2341 *** (0.0399)	−0.1662 *** (0.0275)	−0.1193 *** (0.0310)
*n*	739	739	739
Adj R2	0.0693	0.0822	0.0238

Notes: The dependent variables are robust, engulfed and isolated relations; The results are controlled for the variables presented in Table 4; * *p* < 0.05, ** *p* < 0.01, *** *p* < 0.001.

**Table 4 ijerph-18-05135-t004:** Relationship between student’s behavior and students’ relations.

	Secure	Insecure	
	Robust	Engulfed	Isolated
Internal behavior	−0.1902 *** (0.0382)	−0.0468 (0.0563)	0.0993 ** (0.0499)
External behavior	−0.0293 (0.0377)	0.0994 ** (0.0545)	0.0398 (0.0485)

Notes: The dependent variable is internal behavior (binge drinking) and external behavior (poor body self-esteem); The results are controlled for the variables presented in Table 4; ** *p* < 0.01, *** *p* < 0.001.

## Data Availability

The datasets analyzed in this study are not publicly available but are available on request from the corresponding author.

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
