# Peer review of "Does It Run in the Family? How Family Background Affects Attachment Styles for Students in Higher Education"

_ijerph, 2021, doi:10.3390/ijerph18105135_

Round 1
Reviewer 1 Report
The manuscript is exemplary! Only the heading "Limitation" is without numbering.
Author Response
The heading "Limitation" is now numbering (in blue color in the manuscript): 4.1. Limitations
Reviewer 2 Report
The study carried out by authors was very interesting and original.
There are some points that could be improved.
- In section Material and Methods:
The data collection was done at the educational programs. What criterion did the authors use to include students to the study and to choose exactly these countries: Denmark, Germany, Portugal, Croatia and the Czech Republic?
Was the questionnaire validated in the native language of the students. If no, was it conducted in original language and was the skill level of the participants sufficient to correctly understand the questions in the questionnaire?
Did the authors calculate estimated beta for the study? Did the number of enrolled patients let to reach enough statistical power?
“As our missing information is mainly driven by Germany, where the sample is very large, it will not bias the results substantially” – did the authors calculate it somehow?
Why the authors choose the a simple least square regression to data analysis? It seems that this method is used to linear relations, but the authors categorized social attachment into robust, engulfed and isolated relations and independent variables also were categorized. Use of this method requires some conditions, for example normal distribution of residuals - have the authors checked these conditions?
The one of proper analysis used for assessing relation between categorized variables is logistic regression model with OR calculation.
Authors did not include any information about software program using to data statistical analysis.
There is no information about number of students from different field of study: humanities and social sciences. There should be also specified number of subgroups: robust, engulfed and isolated relations.
The authors mentioned about BMI and Body image, but there were no analysis with these factors in section of Results.
- In section Results:
Table 1. Which was the meaning of values in column „mean, SD”? In characteristics table there is no numbers, percentages of students included in individual categories like: with high income, high education, etc.
In table 2. it is not clear which number of group was included to analysis, for example: in subgroup engulfed n=155 or 98??
Due to categorized character of data, it seems to be more adequate using chi-square test to estimate differences between subgroups robust, engulfed and isolated due to family background. This test is used to compare proportions between subgroups.
In Table 2-4 the results were not clearly presented. Which was the meaning of values: coefficient or likelihood or percentage?
Why did not the authors include to the analysis all factors like fulltime working or firstborn?
Author Response
We are grateful, the comments improved greatly the quality of the manuscript content and were all introduced in the text accordingly (in blue colour). We hope that this revised manuscript will satisfy the main points stated.

Reviewer 3 Report
I underline the main positive aspects of the paper.
- The analysis of the problem in the light of the post-modern landscape (or late modernity): this approach allows to understand the issue by combining statistical data and philosophical glance.
- The good theoretical and conceptual framework (including the attention on the religious factors for understanding the social transformations).
- The presentation of the critical tools (materials and methods) is good.
- The transition from the relevance of the traditional social class to the increasing role of the family characteristics is very interesting.
- The nexus between the economic capital and social capital: a typical trait of the post-modern process of social assimilation.
- The critical approach of the authors to the results of the analysis (i.e. endogeneity): they are conscious about the necessity to enrich the research with other studies.
Conclusion. Notwithstanding the short extension, in my opinion the paper provides a good and well-documented overview about the composition of the social factors underlying the formation/education within complex and post-modern societies.
Author Response
Thank you for your comments concerning our manuscript. Those comments are all valuable and important to continue our research focus.
Reviewer 4 Report
The article addresses the topic of current relevance. Indeed, over the last years, researchers increasingly paid attention to the impact of family structure on the child’s future. The authors provided an extensive review of scientific works. The theory is well written. The text enjoys a good writing style and accuracy of presentation.
Questions to the authors:
- Have you tried to identify the distinctions in the style of attachment in students from different countries?
- Is there a linkage between family background and the academic achievement of students?
Author Response
Thank you for your comments concerning our manuscript. Your questions are all valuable and important to continue our research focus because we do not yet have the answers for your questions.